# The Utilization and Impact of Dopamine Transporter Imaging in Diagnosing Movement Disorders at a Tertiary Care Hospital in Greece

**DOI:** 10.3390/biomedicines13040970

**Published:** 2025-04-16

**Authors:** Georgia Xiromerisiou, Iro Boura, Eleni Barmpounaki, Panagiotis Georgoulias, Efthimios Dardiotis, Cleanthe Spanaki, Varvara Valotassiou

**Affiliations:** 1School of Medicine, University of Thessaly, 41500 Larisa, Greece; 2School of Medicine, University of Crete, 70013 Heraklion, Greece; boura.iro@gmail.com (I.B.); elenibarb2000@gmail.com (E.B.); 3Department of Nuclear Medicine, University General Hospital of Larisa, Faculty of Medicine, University of Thessaly, 41500 Larisa, Greece; pgeorgoul@uth.gr (P.G.); vvalotasiou@gmail.com (V.V.); 4Department of Neurology, University General Hospital of Larisa, Faculty of Medicine, School of Health Sciences, University of Thessaly, 41500 Larisa, Greece; edar@med.uth.gr

**Keywords:** movement disorders, dopaminergic imaging, DaTscan, Parkinson’s disease, tremor

## Abstract

**Background/Objectives**: The introduction of dopamine transporter scan (DaTscan) in clinical diagnostics has revolutionized the way clinicians approach movement disorders, offering valuable insights into presynaptic striatal dopaminergic deficits and revealing subjacent neurodegeneration. The aim of our study was to evaluate the impact of DaTscan on diagnostic decisions regarding movement disorders, particularly Parkinson’s disease (PD) and atypical parkinsonian syndromes, under real-world circumstances in Greece. **Methods**: We retrospectively analyzed data from 360 patients who underwent a DaTscan examination between 2018 and 2023 at a tertiary hospital in Greece, including referrals from both movement disorder specialists and general neurologists, either hospital-based or in private practice. Demographics, primary referral symptoms, and both pre-scan and post-scan diagnoses were collected and analyzed. **Results**: The mean age in our cohort was 60 ± 13.5 years, and tremor was the leading referral symptom (40.8%). The initial diagnosis changed in nearly half of the cases (48.3%) following DaTscan. Significant shifts included transitions from an “Unclear” or “Dystonia” diagnosis to “Parkinson’s disease” in 78.1% and 72.7% of patients, respectively. However, the particularly high concordance rates between pre-scan and post-scan diagnosis for “Vascular parkinsonism” (100%), “Parkinson’s disease” (89.3%), and “Essential/Dystonic Tremor” (86%) suggest that the test may have been over-utilized or ordered beyond its intended indications. **Conclusions**: DaTscan markedly enhances diagnostic accuracy for movement disorders, particularly for general neurologists, addressing the complexities of overlapping clinical presentations. Continuous medical training is essential to ensure the cost-effective utilization of DaTscan in routine clinical practice; ongoing technological advancements will further refine and expand their applications, benefiting both patients and the broader medical community.

## 1. Introduction

The journey of dopamine transporter scan (DaT, DaTscan) in the medical landscape has been transformative. From its initial introduction to its current widespread acceptance in research and clinical practice, DaTscan has reshaped the way clinicians approach and manage movement disorders [1]. Its value extends beyond the diagnostic potential, impacting patients’ care and adding to the body of neurological research.

Historically, neurologists have grappled with the inherent challenges of diagnosing movement disorders, hindered by overlapping symptoms [2]. In the era preceding advanced imaging techniques, clinicians had to rely heavily on symptomatic presentations, which, although crucial, did not always convey a complete or accurate picture. Misinterpretations and, consequently, misdiagnoses were not uncommon, leading to treatment delays or even interventions not optimally tailored to patients’ needs.

The emergence of DaTscan marked a paradigm shift. Through visualizing presynaptic dopamine reuptake transporters in the brain, this innovative technology provided insight into neural pathways and their functioning [3]. This was groundbreaking for conditions like Parkinson’s disease (PD), which is characterized by the hallmark degeneration of nigrostriatal dopaminergic neurons. The ability to objectively observe such deficits, often before they manifest clinically, has assisted clinicians in diagnostic dilemmas, including distinguishing prodromal PD individuals in at-risk populations [4]. In addition to being a diagnostic biomarker, DaTscan can serve as a dynamic tool for monitoring disease progression and assessing intervention outcomes [5]. Such versatile applications underscore its central role in current and future patient management.

On a broader scale, DaTscan has catalyzed research in neurodegenerative diseases [6]. By offering tangible, visual evidence of neural changes which correlate to clinical manifestations, it has enabled a deeper exploration into the pathophysiological mechanisms behind disorders like PD [7,8]. This richer, more nuanced understanding holds promise for the development of novel therapeutic interventions and, potentially, preventive strategies; this explains the incorporation of DaTscan, as an index examination of subjacent neurodegeneration, in the recently suggested biological classification schemes for PD [9].

The aim of the present study was to evaluate the impact of DaTscan results on diagnostic decisions in a cohort of neurological patients with movement disorders under real-world conditions. Patients referred by movement disorder specialists or general neurologists, working either in hospital settings or in private practice, underwent a DaTscan examination at a tertiary hospital in Greece.

## 2. Materials and Methods

We retrospectively collected demographic information from all adult patients who underwent a DaTscan between 2018 and 2023 in the Nuclear Medicine Laboratory of a tertiary hospital in Greece. The included patients were referred by either the Movement Disorders Outpatient Clinic of our hospital or by private practice general neurologists. For each patient, their primary referral or dominant symptom was documented, along with their presumed diagnosis before and after undergoing a DaTscan examination. Treating neurologists were specifically asked whether the DaTscan results confirmed their initial suspicion or led to an alternative diagnosis. All patients underwent computed tomography (CT) and/or magnetic resonance imaging (MRI) prior to the DATscan. Patients with structural brain lesions were excluded.

All patients underwent a DaTscan single photon emission computed tomography (SPECT) 3–4 h after the intravenous (IV) administration of 185 MBq I-123-ioflupane. Drugs that interfere with tracer uptake were discontinued according to the European Association of Nuclear Medicine (EANM) guidelines [10,11]. Oral iodine solution was given for thyroid protection. SPECT scans were performed on a dual-head γ camera (Infinia, GE Healthcare) in a step-and-shoot mode (angular step 3°, 40 s/step, rotation coverage 360°), using low-energy high-resolution parallel-hole collimators and a matrix size of 128 × 128. The rotational radius was set at 15 cm and the photopeak was centered at 159 KeV ± 10%. Image reconstruction was carried out using iterative algorithms (ordered subset expectation maximization, OSEM). We applied Chang’s attenuation correction and Butterworth filtering. No scatter correction was made.

DaTscan SPECT studies were independently evaluated both visually and semi-quantitatively by two experienced nuclear medicine physicians. Images of the whole right (R) and left (L) striatum, R-L putamen and R-L caudate were rated visually as normal or abnormal/reduced uptake. A semi-quantitative evaluation was performed to assess specific DaT binding by selecting the three transverse slices with the most intense tracer uptake and drawing standardized anatomical regions of interest (ROIs) over the L-R striatum, L-R putamen, L-R caudate and occipital cortex, as the reference background region. Moreover, we performed automated semi-quantification using a commercial software (DaTQUANT™, GE Healthcare, Chicago, IL, USA, https://www.gehealthcare.com/, accessed on 13 April 2025) for the evaluation of percent deviation (given as z-score) of the whole striatum and the substriatal regions, using the means from age-matched normal subjects taken from the Parkinson’s Progression Markers Initiative (PPMI) database (http://www.ppmi-info.org, accessed on 25 October 2023).

The data were subjected to comprehensive analysis, involving both statistical evaluations and visual representations, using R and RStudio (R version 4.4.3) [12,13]. A multiple linear regression model was used to explore the effect size of different variables on the interval between symptom onset and DaTscan referral. A logistic regression model was used to explore the impact size of various factors on the likelihood of a change in diagnosis following DaTscan (binary outcome). For these purposes, individuals with a diagnosis of “Essential/Dystonic tremor” and “Dystonia” were grouped together (“Essential tremor/Dystonia”). Observations with a pre-scan or post-scan diagnosis of “Vascular parkinsonism” were removed for the logistic regression model and for the calculation of the diagnostic performance metrics of DaTscan. Cross-tabulations were generated to compare pre- and post-scan diagnoses. Visualizations were created to convey the distribution of various diagnoses, age groups, and primary symptoms.

## 3. Results

A total of 360 patients were included, with men and women being represented in comparable proportions (51.4% male). Their mean age was 60.0 ± 13.5 years (range: 24–87 years), while they were referred for a DaTscan examination at an average of 2.0 ± 2.7 years after their symptom onset (Table 1). Individuals with an initially “Unclear” diagnosis were referred for a DaTscan earlier than those with a more specific diagnosis. After adjusting for sex and age at the time of enrollment (i.e., the time of DaTscan performance), the interval between symptom onset and DaTscan referral was significantly longer for individuals with a primary referring symptom of dystonia (β = 1.99, SE = 0.7, *p* = 0.001), those with an initial pre-scan diagnosis of “Essential tremor/Dystonia” (β = 1.48, SE = 0.5, *p* = 0.001), and those whose diagnosis did not change following a DaTscan (β = 1.32, SE = 0.3, *p* < 0.001) (Appendix A).

Tremor emerged as the most prevalent referral symptom (40.8%) in our cohort, followed by rigidity (21.4%) and postural instability (20%) (Figure 1).

Potential pre-scan diagnoses included “Essential/Dystonic tremor”, “Parkinson’s disease”, “Vascular parkinsonism”, “Parkinson plus/Atypical parkinsonism” (to signify other neurodegenerative cases of parkinsonism), “Non-neurodegenerative parkinsonism” (other than vascular parkinsonism), and “Dystonia”. With the exception of dystonia and postural instability, where the referral symptoms were strongly associated with the pre-scan diagnosis of dystonia and atypical parkinsonism, respectively, significant heterogeneity was observed in the remaining symptoms and their corresponding pre-scan diagnosis (Figure 2).

The DaTscan SPECT acquisition and reconstruction parameters were similar between our patients and the PPMI control group. The DaTscan results were classified as abnormal (71.1%), normal (26.1%), or undetermined (2.8%) (Figure 3), and led to a change in original diagnosis for nearly half of the referred patients (48.3%) (Figure 4 and Figure 5).

The agreement rate between pre-scan and post-scan diagnosis varied. Patients initially diagnosed with “Vascular parkinsonism”, “Parkinson’s disease”, or “Essential/Dystonic tremor” largely retained their original diagnosis in percentages of 100%, 89.3%, and 86%, respectively. On the other hand, patients initially diagnosed with “Dystonia” were the least likely to keep their pre-scan diagnosis, as it changed to “Parkinson’s disease” in almost three-quarters of cases (72.7%).

The majority of changed diagnoses overall were to “Parkinson’s disease” (69.0%). This trend was significantly driven by patients initially characterized as having an “Unclear” diagnosis, which became more definitive following a DaTscan (95.9%). The newly acquired diagnosis was mostly “Parkinson’s disease” (81.4%). A lower percentage of patients changed their diagnoses to “Vascular parkinsonism” (12.6%), “Parkinson plus/Atypical parkinsonism” (5.2%), and “Essential/Dystonic tremor” (5.2%). Regarding PD patients in particular, a small proportion (4%) transitioned to “Parkinson plus/Atypical parkinsonism” syndromes. Interestingly, 7.5% of patients with an initial specific diagnosis shifted to “Unclear” following the DaTscan result. This percentage comprised patients originally diagnosed with “Parkinson plus/Atypical parkinsonism” or “Parkinson’s disease”, the majority of whom had a normal DaTscan result (scan without evidence of dopaminergic deficit (SWEDD)), though not in all cases. Moreover, there were four cases of “Parkinson plus/Atypical parkinsonism” or “Parkinson’s disease” whose diagnosis remained unchanged, despite a normal DaTscan result.

We fitted a logistic regression model to better characterize the factors associated with a change in diagnosis following DaTscan (Appendix A). Patients with a pre-scan diagnosis of “Parkinson’s disease” were significantly less likely to have their diagnosis changed than those with an “Unclear diagnosis” (odds ratio, OR = 0.0015, *p* < 0.001). Similar trends in the odds of diagnosis change were observed for patients with a pre-scan diagnosis of “Essential tremor/Dystonia” (OR = 0.055, *p* < 0.001), “Parkinson plus/Atypical parkinsonism” (OR = 0.006, *p* < 0.001), and “Non-neurodegenerative parkinsonism” (OR = 0.055, *p* < 0.001). Moreover, an abnormal DaTscan result was associated with a nearly sevenfold increase in the likelihood of diagnosis change (OR = 6.74, *p* < 0.001). Notably, each additional year between symptom onset and DaTscan referral reduced the odds of diagnostic revision by 28% (OR = 0.72, *p* = 0.033), suggesting that earlier referrals were associated with greater diagnostic uncertainty.

Finally, we assessed the diagnostic performance of DaTscan using our cohort as reference (Table 2). These metrics were derived based on the concordance between the DaTscan result and the change or stability of diagnosis following the test, and were found to be in line with previously reported meta-analyses differentiating neurodegenerative from non-neurodegenerative forms of parkinsonism [1].

## 4. Discussion

The integration of modern imaging techniques into clinical diagnostics has revolutionized the approach to complex neurological disorders. Available data from the global medical community supports the role of DaTscan as a valuable asset in the diagnosis and management of movement disorders, where clinical presentations are often ambiguous and an accurate diagnosis may be challenging [14]. Our study emphasizes the transformative impact of DaTscan within the intricate field of movement disorders, highlighting the indispensable role of clinical acumen not only in understanding clinical manifestations and DaTscan indications, but also in accurately interpreting DaTscan results, a task particularly challenging for less-experienced clinicians. To our knowledge, this is the largest study to explore the role of DaTscan in distinguishing between various movement disorders in Greece, a country with a mixed public–private healthcare system significantly affected by the financial crisis. DaTscan imaging is routinely used in tertiary healthcare facilities in Greece, with past studies demonstrating its utility in differentiating between PD from non-degenerative parkinsonism and essential tremor (ET) [15,16,17].

Our cohort predominantly consisted of patients in their senior years, reflecting the typical onset age for many movement disorders [18,19]. Interestingly, patients whose primary referral symptom was dystonia tended to be younger, yet their referral for a DaTscan examination appeared to be significantly delayed. The prolonged interval between symptom onset and DaTscan referral may reflect either a reluctance from patients to seek medical attention promptly or hesitation from clinicians to reconsider their initial diagnosis. Notably, this interval was longer among patients whose diagnosis remained unchanged following DaTscan and was significantly prolonged in cases with a pre-scan diagnosis of either ET or dystonia. This pattern suggests that DaTscan imaging may have been employed later in the disease course, serving more as a confirmatory test rather than an early diagnostic tool. Such use could reflect clinicians’ uncertainty in the presence of atypical clinical features, often observed in the course of movement disorders, including the controversial ET-plus cases [20]. Alternatively, it may indicate patients’ reluctance to accept their initial diagnosis or their dissatisfaction with the long-term response to treatment or drug-induced side effects, particularly among younger individuals with ET and a greater need for effective symptom control [21]. Nonetheless, this finding raises questions regarding the clinical utility of late-stage DaTscan referrals, especially in cases where diagnostic certainty is already high.

Few studies have chosen to include referral symptoms as well as initial diagnoses of patients undergoing a DaTscan examination [22,23]. Similarly to them, tremor was the leading symptom prompting a DaTscan recommendation in our cohort. This observation is indicative of the diagnostic challenges faced by neurologists, as tremor is present in a plethora of prevalent conditions, including PD, ET, dystonia, and even drug-induced disorders. The frequent occurrence of tremor in approximately two-fifths of all DaTscan referrals in our cohort highlights clinicians’ reduced diagnostic confidence in addressing this symptom, alongside the potential for misdiagnosis. Rigidity, the second most common referral symptom, further complicates the diagnostic conundrum due to being present in various neurodegenerative and non-neurodegenerative conditions, but also in the aging population [24]. However, none of the referral symptoms per se were found to impact the likelihood of a change in diagnosis following DaTscan.

The high proportion of abnormal DaTscan results in our cohort (73.9%), indicative of presynaptic DaT density reduction, showcases the value of this test in confirming or refuting the clinical suspicion of subjacent neurodegenerative parkinsonism. Notably, an abnormal DaTscan result was significantly associated with a change in diagnosis, particularly among cases with an initially uncertain diagnosis, the majority of whom were ultimately diagnosed with PD. Overall, the shift in post-scan diagnosis in almost half of our patients agrees with the percentages reported by other studies across the globe [1], serving as a testament to the DaTscan impact on the clinical diagnostic process. Indeed, the transition of almost all patients (95.6%) classified as “Unclear” to a more definitive post-scan diagnosis reinforces the utility of DaTscan in routine clinical practice.

Intriguing insights can be derived from reviewing the percentages of diagnoses that were either revised or remained unchanged following DaTscan. An initial diagnosis of “Vascular parkinsonism” was largely retained, showing high confidence of clinicians in diagnosing this condition. This finding could possibly reflect the distinct clinical features of this entity (e.g., cardiovascular risk factors, stroke history, stepwise progression, lower body parkinsonism, no tremor, etc.), but also the coexistence of specific CT and/or MRI findings, which provide a more objective basis and are less reliant on the clinician’s experience and expertise [25]. This observed trend of diagnostic stability was not reflected in our logistic regression model, likely due to the large standard error stemming from the small number of patients in this diagnostic category. As a result, we decided to exclude this category from the final model. However, we highlight this observation, as it reflects real-world clinical practice, while the limited number of DaTscan referrals for patients with suspected vascular parkinsonism may, in itself, suggest that clinicians often consider the diagnosis sufficiently certain to not rely on DaTscan results. On the other hand, approximately 12% of the patients whose diagnosis changed following a DaTscan were classified as having “Vascular parkinsonism”. DaTscan results vary significantly across vascular parkinsonism cases with both normal and abnormal findings reported [26], a pattern also observed in our cohort. Though the shift in diagnosis cannot be solely explained by this crude classification of DaTscan imaging, other parameters not included in our data, such as the unusual pattern of the reduced striatal uptake of the tracer (e.g., patchy or highly asymmetrical, unilateral involvement of both the caudate and putamen), MRI findings, or cardiovascular risk factors, could have contributed as well [27]. Moreover, the reasons behind a DaTscan request in patients with possible vascular parkinsonism are not solely diagnostic; therapeutic implications are also acknowledged, as an abnormal DaTscan result, typical for PD, might imply a comorbidity of the two conditions, and has been associated with a better response to levodopa, thereby affecting patients’ care [28].

For patients initially diagnosed with “Parkinson’s disease”, DaTscan solidified this diagnosis, aligning with the high sensitivity of DaT imaging in autopsy-confirmed PD cases [29]. However, given the results of a large meta-analysis showing pooled diagnostic accuracies of almost 74% for non-experts and 80% for movement disorder specialists in clinically diagnosing PD [30], the percentage of nearly 90% of unchanged diagnoses among patients initially diagnosed with PD in our cohort appears higher than anticipated. PD diagnosis is considered to be primarily clinical-based, with DaTscan referrals kept mostly for cases with atypical or subtle presentations and an uncertain diagnosis [6]. Consequently, this high percentage could be indicative of inappropriate DaTscan referrals. This trend was previously observed in almost 40% of DaTscan requests in a community-based hospital (referrals from both general neurologists and movement disorders specialists) and was associated with an absence of practical benefits in patient management, thus undermining the cost-effectiveness of the test in routine clinical practice [31]. The presence of non-hospital-based general neurologists among the referring clinicians may account for this finding in our cohort as well, as such misconceptions are more prevalent in this group compared to movement disorder experts [32]. The misdiagnosis of ET is also common in routine clinical practice; previous studies have reported that nearly half of the cases initially determined as ET were later reassigned to a different neurological condition, mostly dystonia or PD [33,34]. However, this trend was not confirmed in our cohort, as the high agreement percentage (86%) between the pre- and post-scan diagnoses of ET may reflect this same notion of potentially inappropriate DatTscan requests. These findings underscore the importance of constant education within the medical community, with a special focus on widely available, advanced diagnostic tools.

Notably, there were five cases in which the diagnosis changed from “Parkinson’s disease” to “Parkinson plus/Atypical parkinsonism” or vice versa. The DaTscan result was abnormal in all instances, suggesting that it was not the sole determinant of the diagnostic decision and that other factors beyond the scope of our study may have contributed. Moreover, characterizing a DaTscan result solely as “abnormal” could be an oversimplification. The importance of identifying distinct nigrostriatal uptake patterns in DaT imaging (e.g., symmetrical and dot-shaped instead of the typical-for-PD unilateral comma-shaped pattern of tracer uptake in DaTscan imaging), which could assist in distinguishing PD from specific atypical parkinsonian syndromes, such as multiple system atrophy (MSA), has been prominently emphasized within the research and medical community [6,11]. Moreover, though the accuracy of DaTscan is compromised in differentiating PD from atypical parkinsonian syndromes [35], their combination with other diagnostic tools, such as I-123 meta-iodobenzylguanidine (MIBG) scintigraphy, I-123 iodobenzamide (IBZM) dopamine receptor scintigraphy, or urine residual volume, could further assist in diagnostic dilemmas [17,36].

The post-scan diagnosis changed to “Parkinson’s disease” in almost three-quarters of those referred for DaTscan with an initial diagnosis of “Dystonia”, with all changes driven by abnormal DaTscan results. In our regression analysis, we combined the “Dystonia” and “Essential/Dystonic tremor” groups to reduce error introduced by the small sample size of the former. This grouping was supported by comparable age distributions and conceptual clinical overlap [37] between dystonia and tremor cases referred for DaTscan imaging, allowing for broader trend observations within the cohort. Nonetheless, we highlight some distinct outcomes in the “Dystonia” group, as they offer meaningful insights into clinical practice. More specifically, dystonia in PD is usually levodopa-induced, occurring as a wearing-off manifestation in advanced PD, a peak dose phenomenon, or in the context of biphasic dyskinesia [38]. The younger age at symptom onset in this subpopulation in our cohort (Table 1) may signify PD cases with dystonia as a presenting or an early sign in the disease course, as such manifestations are frequently observed in early-onset or genetic forms of PD [38]. Since dystonia is not typically described in PD, a high proportion of DaTscan requests in this context would be expected. However, in our cohort, dystonia as the primary referral symptom was associated with a DaTscan referral later in the disease course. Due to the small sample size of this patients’ group with heterogeneous diagnoses, larger studies are needed to further investigate this association.

Despite its clinical utility, also confirmed by the high overall diagnostic accuracy found in our cohort, DaTscan is not without limitations. In non-neurodegenerative parkinsonism, false positive results may occur in cases of inappropriate patient preparation, such as the non-discontinuation of medicines and other substances which interact with tracer uptake [39], due to prior head trauma or other basal ganglia pathology [40]. Nevertheless, in our cohort, special care was taken to exclude patients with brain lesions that could confound the DaTscan results. Additionally, the Nuclear Medicine team at the tertiary hospital comprised experienced personnel who provided proper guidance to patients regarding the preparation for the test, including instructions on medications to avoid prior to the procedure. False negative results may be attributed to lack of experience, especially in early-stage neurodegeneration with minimal impairment of nigrostriatal pathway, or due to late-onset PD with suboptimal striatal adaptive mechanisms, as recently suggested [41]. Follow-up studies of patients with suspected PD and normal DaTscan at baseline are conflicted, reporting that 12.5% to 65.6% of these patients did have abnormal DaTscan results in the long-term [42,43]. However, larger longitudinal studies detected much smaller percentages of reduced nigrostriatal uptake in repeated DaTscan tests (up to 3.6%), with researchers strongly pointing towards the direction of non-neurodegenerative underlying disorders [22,44]. Moreover, normal DaTscan results were reported in selected patients with presumably presynaptic dopaminergic parkinsonism, such as pathologically confirmed cases of corticobasal degeneration (CBD), progressive supranuclear palsy (PSP), or MSA [29]. These findings may explain the relatively low negative predictive value in our cohort and the percentage of patients with a normal DaTscan result who maintained a post-scan diagnosis of neurodegenerative parkinsonism.

Methodological limitations are present in our study. The retrospective approach we followed introduces the possibility of confounding, as unmeasured risk factors may have influenced our findings. Additionally, real-world studies, like ours, often lack quality control in data collection and are prone to various bias sources when comparing outcomes [45]. Furthermore, our study does not include long-term follow-up visits for our patients, which could have provided further confirmation or adjustment of post-scan diagnoses. On the other hand, community-based studies are valued and insightful, as they reflect clinical care practices among unselected patients and clinicians. They provide critical information on real-world circumstances, including shedding light on administrative and logistical problems, which are typically waived in controlled research settings. While such studies are indispensable for guiding healthcare policies, their methodological challenges must be carefully considered.

Our data underscores that DaTscan is most beneficial for patients with an unclear initial diagnosis. Although it should always be considered in the context of individual patients’ symptoms and particularities, DaTscan can assist in differentiating neurodegenerative from non-neurodegenerative parkinsonian syndromes, thereby aiding clinical decision-making and informing prognosis [46]. On the other hand, a thorough understanding of the role of DaTscan in clinical diagnostics, including appropriate indications for referral, the importance of prudent and rational application, and the need to maintain realistic expectations regarding its capabilities, is necessary. Based on our findings, we suggest that the timing and context of DaTscan referral are critically linked to its clinical utility. Referrals made late in the disease course, particularly when a diagnosis has already been established, may offer limited practical value for patient management. This appears especially relevant for individuals with a longstanding diagnosis of ET, unless there is substantial clinical suspicion of an ET-to-PD phenotypic shift, a possibility supported by previous literature [47]. Additionally, DaTscan does not appear to play a significant role in revising a diagnosis of vascular parkinsonism, as this entity is typically supported by robust clinical and imaging findings. However, in cases where comorbid PD is suspected, a DaTscan may still provide clinically relevant information. Comprehensive training for clinicians is essential to leverage the full potential of DaTscan, while individual patient factors, such as comorbidities, medications, and age, should be carefully considered when interpreting the results. Although DaTscan has been criticized as an expensive examination, not widely available in routine clinical practice [48], a judicious use could decrease unnecessary medication administration and complex or time-consuming additional tests, thus improving patients’ outcomes and reducing healthcare costs in the long term [49]. Larger studies with a greater number of individuals in certain categories, such as those diagnosed with dystonia, are needed. Additionally, more comprehensive clinical data surrounding the reasons for these DaTscan referrals, along with extended follow-up observation periods to validate diagnosis, are essential to further corroborate these findings.

## 5. Conclusions

Our study at a tertiary hospital in Greece highlights the multi-faceted and pivotal role of DaTscan examinations in assisting clinicians to address the complexities of movement disorders diagnosis. The visual representations of our results underscore their impact on diagnostic accuracy, with notable shifts from specific pre-scan to post-scan diagnoses. Continuous technological advancements will further refine DaTscan utility and expand their role in neurology, benefiting both patients and the broader medical community. The symbiotic relationship between clinical expertise and technological innovation will undoubtedly shape the future of patient care, offering hope and precision in the ever-evolving landscape of neurology. As the medical community continues to confront the challenges posed by overlapping clinical presentations in movement disorders, modern diagnostic tools like DaTscan stand out, guiding clinicians toward more accurate diagnoses and effective therapeutic interventions.

## Figures and Tables

**Figure 1 biomedicines-13-00970-f001:**
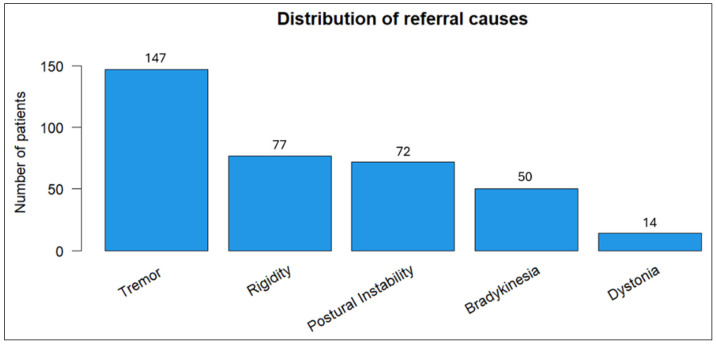
Distribution of primary referral symptoms among patients.

**Figure 2 biomedicines-13-00970-f002:**
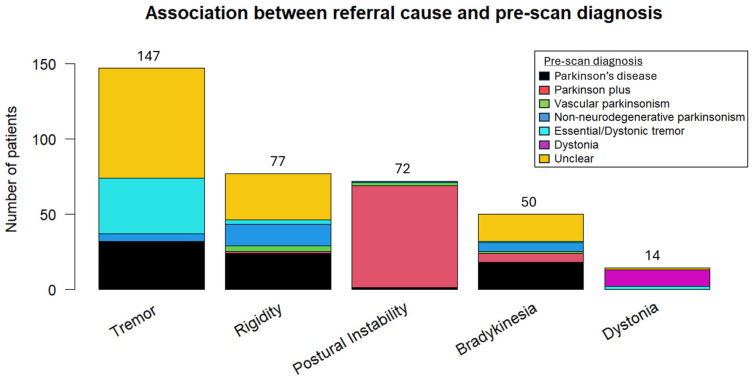
Association between the referral or dominant symptom and pre-scan diagnosis.

**Figure 3 biomedicines-13-00970-f003:**
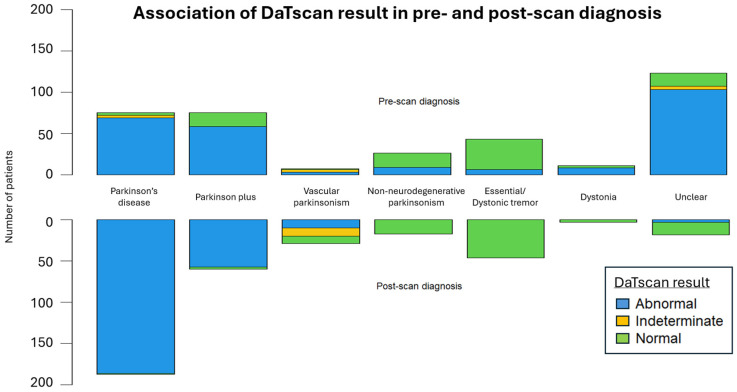
The mirrored stacked bar plots display the distribution of DaTscan results across pre-scan and post-scan diagnoses.

**Figure 4 biomedicines-13-00970-f004:**
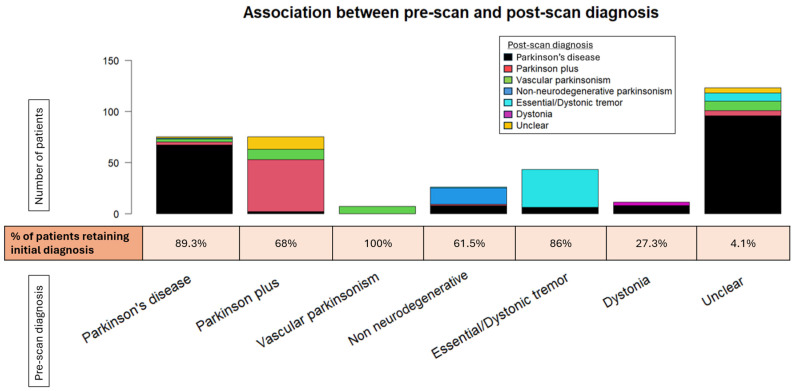
The stacked bar plot displays the shift across the various pre-scan diagnoses. The percentage of patients retaining their original diagnosis is depicted below the bar plot for each given diagnosis.

**Figure 5 biomedicines-13-00970-f005:**
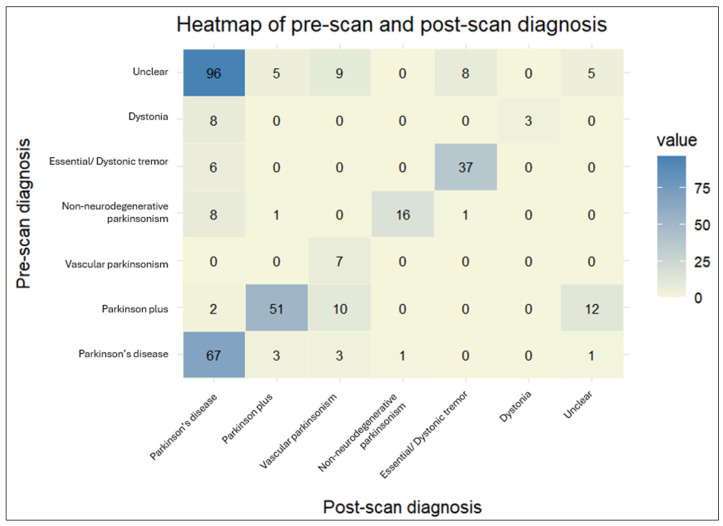
The heatmap provides a visual representation of the cross-tabulated pre- and post-scan diagnoses. The darker shades indicate a higher number of patients for a particular combination of pre- and post-scan diagnoses.

**Table 1 biomedicines-13-00970-t001:** Summary of age metrics in various groups with distinct primary referral symptoms.

	Age at Symptom Onset (y)	Time to DaTscan Referral (After Symptom Onset) (y)
Overall (n = 360)	56.9 ± 10.4	1.5 ± 1.1
Primary referral symptom:		
Tremor (n = 147)	49.7 ± 11.9	2.1 ± 2.4
Rigidity (n = 77)	63.1 ± 9.8	1.6 ± 1.1
Postural instability (n = 72)	71.3 ± 8.0	1.8 ± 3.1
Bradykinesia (n = 50)	60.6 ± 11.8	1.8 ± 1.1
Dystonia (n = 14)	39.1 ± 10.8	5.1 ± 7.5 **
Pre-scan diagnosis		
Parkinson’s disease (n = 75)	62.5 ± 9.1	1.7 ± 1.3
Atypical parkinsonism (n = 75)	70.1 ± 8.6	1.8 ± 3.1
Vascular parkinsonism (n = 7)	74.1 ± 7.3	2.0 ± 2.3
Non-neurodegenerative parkinsonism (n = 26)	56.8 ± 11.8	2.0 ± 0.9
Essential tremor/dystonic tremor (n = 43)	40.3 ± 11.3	3.8 ± 3.7 **
Dystonia (n = 11)	38.6 ± 11.5	5.4 ± 8.3 **
Unclear (n = 123)	54.7 ± 10.9	1.5 ± 1.0
Change in diagnosis		
Yes (n = 174)	57.3 ± 12.0	1.4 ± 0.9 ***
No (n = 186)	58.6 ± 15.7	2.5 ± 3.5 ***

** *p* = 0.001, *** *p* < 0.001. n: sample size; y: years

**Table 2 biomedicines-13-00970-t002:** Diagnostic performance metrics of DaTscan in the study cohort.

Sensitivity	92.7%
Specificity	95.6%
Positive Predictive Value	98.7%
Negative Predictive Value	77.6%
False Positive Rate	4.3%
False Negative Rate	7.2%
Accuracy	93.3%

## Data Availability

The data presented in this study are available on request from the corresponding authors.

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
