# Peer review of "The Utilization and Impact of Dopamine Transporter Imaging in Diagnosing Movement Disorders at a Tertiary Care Hospital in Greece"

_biomedicines, 2025, doi:10.3390/biomedicines13040970_

Round 1

Reviewer 1 Report

Comments and Suggestions for Authors

In this retrospective study, the authors have assessed the impact of Dopamine Transporter scan (DaTscan) results on diagnostic decision in 360 patients with several movement disorders.   Given the high prevalence of movement disorders and the challenges faced routinely by the neurologists when establishing the diagnosis, understanding the role and utility of DaTscan could have significant clinical implications for early and correct detection of these conditions.   

The manuscript is well-organized, the methods and the results are well structured and clearly presented. The conclusions support the main findings of the research, although this section may be improved by the authors to enhance the strengths of their study. The authors have offered updated evidence of current data regarding DaTscan using appropriate references, although the discussions can be extended with more referrals to similar research.  

I have some minor suggestions for the authors.  

  • please add some possible explanations for the variable time to DaTscan referral after symptom onset. Except for the unclear diagnosis from the symptom onset, do the authors have information regarding the reasons why the patients were referred to DaTscan (for instance atypical evolution, suboptimal response to medication, financial reasons, research purposes, etc)
  • the authors have presented in figure 2 the association between the referral symptom and pre-scan diagnosis. Do the authors have an evidence regarding the association between the referral symptom and post-scan diagnosis? If yes, were there notable changes between pre-scan and post-scan diagnosis in relation to the referral symptom?
  • based on the results observed in their cohort (and also considering that DaTscan is not widely available), I suggest to the authors to add a paragraph highlighting the practical relevance of their findings, in regard with some particular indications of DaTscan that should not be missed in clinical care practice and future research (e.g. distonia, which may be an underrecognized symptom of Parkinson's disease). Also, please expand the future research directions section.

Author Response

Reviewer 1

I have some minor suggestions for the authors.  

  • please add some possible explanations for the variable time to DaTscan referral after symptom onset. Except for the unclear diagnosis from the symptom onset, do the authors have information regarding the reasons why the patients were referred to DaTscan (for instance atypical evolution, suboptimal response to medication, financial reasons, research purposes, etc).

We thank the reviewer for this insightful comment, which allowed us to further elaborate on this important aspect. Indeed, the time to DaTscan referral varies considerably among patients. Unfortunately, we don’t have more specific clinical information or the reasons behind the patients’ referrals. To investigate potential trends, we built a multiple linear regression model to assess the effect size of various factors on the interval between symptom onset and referral for DaTscan. After adjusting for sex and age at enrollment (i.e., age at DaTscan performance), we found that dystonia as the primary referral symptom was associated with potential delays in DaTscan referral. Conversely, patients whose diagnosis changed following DaTscan were referred earlier compared to those with unchanged diagnosis, suggesting that clinicians with strong diagnostic uncertainly may expedited the use of DaTscan to support clinical decision-making. A pre-scan diagnosis of essential/dystonic tremor or dystonia was associated with longer intervals between symptom onset and DaTscan referrals. These findings suggest that in such cases, DaTscan may have been employed later in the disease course as a confirmatory test rather than an early diagnostic tool. This raises questions about the clinical utility of late-stage DaTscan referrals in cases where diagnostic certainty is already high.

Although the regression model demonstrated a relatively low adjusted R-squared value (0.13), indicating limited predictive power and the influence of other unmeasured factors, we believe the effect sizes of the aforementioned variables are informative, while the statistically significant associations for the pre-scan diagnoses of essential/dystonic tremor and dystonia provide clinically meaningful insights. Therefore, our interpretation focused on understanding effect sizes and exploring patterns in clinical practice rather than making predictions. In accordance with the above, we have updated the Results and Discussion section (highlighted in yellow). Moreover, we have modified Table 1 to include the intervals between symptom onset and DaTscan referrals across different patients’ groups.

  • the authors have presented in figure 2 the association between the referral symptom and pre-scan diagnosis. Do the authors have any evidence regarding the association between the referral symptom and post-scan diagnosis? If yes, were there notable changes between pre-scan and post-scan diagnosis in relation to the referral symptom?

This is an interesting observation and we thank the Reviewer for raising this issue. After fitting a logistic regression model in our data we found that the referral symptoms did not have a significant effect on the likelihood of a change in diagnosis following DaTscan. We have updated the Results section accordingly.

  • based on the results observed in their cohort (and also considering that DaTscan is not widely available), I suggest to the authors to add a paragraph highlighting the practical relevance of their findings, in regard with some particular indications of DaTscan that should not be missed in clinical care practice and future research (e.g. dystonia, which may be an underrecognized symptom of Parkinson's disease). Also, please expand the future research directions section.

We thank the Reviewer for this valuable suggestion. We have added an extra paragraph in the end of the Discussion section summarizing some trends detected in our findings and highlighting their potential association to clinical practice (lines 380 – 389). Moreover, we have expanded on the section of future directions (lines 395 – 400).

Reviewer 2 Report

Comments and Suggestions for Authors

Dear authors

The manuscript brings important insights to the area of ​​diagnostic imaging in neurological diseases, focusing on Parkinson's disease and Parkinsonism and movement disorders. DaTscan is a paradigm shift that helps visualize dopaminergic reuptake via transporter in presynaptic neurons, and with this we can have a functional view of the brain architecture.
The sample was satisfactory for using inferential power. I congratulate the authors for the excellent sample included in the study. In addition, the inclusion of future implications for the use of DATscan in the diagnostic part was appreciable.
The agreement was high between “Vascular Parkinsonism” and “Parkinson's disease before and after DATscan, which reinforces the performance of the diagnostic evaluation system. The exemption of the ICC analysis may be appreciable. In the analysis, I would like to see all the values ​​of false positives, false negatives, positive and negative predictive values, accuracy and diagnostic sensitivity for the use of DATscan.

I would like to see regression analyses in order to conclude the possible cause/effect relationship.

The analysis of a single point in time limits long-term inferences and does not leave inferential power for post-examinations, however, it provides crucial information on clinical practice and plausible interpretations with real-world decision-making.

The manuscript is important for the area and has merit, it will be of great value to the area of ​​diagnosis of neurological diseases and movement disorders.

Author Response

Reviewer 2

The agreement was high between “Vascular Parkinsonism” and “Parkinson's disease before and after DATscan, which reinforces the performance of the diagnostic evaluation system. The exemption of the ICC analysis may be appreciable. In the analysis, I would like to see all the values ​​of false positives, false negatives, positive and negative predictive values, accuracy and diagnostic sensitivity for the use of DATscan.

We thank the Reviewer for suggesting this valuable addition. We have calculated the diagnostic performance metrics of DaTscan in our cohort and presented them in Table 2 in the Results section. However, regarding vascular parkinsonism in particular, we decided to remove these observations from the calculations, as the result of DaTscan in these cases could be either positive or negative and from the available information of our dataset we were not able to conclude in which cases this result was informative or not. We have highlighted this limitation in our Discussion section (lines 264 – 270).

I would like to see regression analyses in order to conclude the possible cause/effect relationship.

We would like to thank the Reviewer for the opportunity to refine our analysis and strengthen our conclusions. In response to their suggestion, we conducted two regression analyses: a multiple linear regression to evaluate the effect of available factors on the interval between symptom onset and DaTscan referral, as recommended by Reviewer 1, and a logistic regression model to assess how these factors might influence the likelihood of a change in the initial pre-scan diagnosis following DaTscan. The Methodology and Results sections have been updated accordingly, plus we have included the detailed results of our analysis as Supplementary Material for your review.
